# VoxGRAF: Fast 3D-Aware Image Synthesis with Sparse Voxel Grids

Katja Schwarz[1]    Axel Sauer[1]    Michael Niemeyer[1]    Yiyi Liao[2]    Andreas Geiger[1]

[1]University of Tübingen and Max Planck Institute for Intelligent Systems, Tübingen
[2] Zhejiang University, China

## Abstract

State-of-the-art 3D-aware generative models rely on coordinate-based MLPs to parameterize 3D radiance fields. While demonstrating impressive results, querying an MLP for every sample along each ray leads to slow rendering. Therefore, existing approaches often render low-resolution feature maps and process them with an upsampling network to obtain the final image. Albeit efficient, neural rendering often entangles viewpoint and content such that changing the camera pose results in unwanted changes of geometry or appearance. Motivated by recent results in voxel-based novel view synthesis, we investigate the utility of sparse voxel grid representations for fast and 3D-consistent generative modeling in this paper. Our results demonstrate that monolithic MLPs can indeed be replaced by 3D convolutions when combining sparse voxel grids with progressive growing, free space pruning and appropriate regularization. To obtain a compact representation of the scene and allow for scaling to higher voxel resolutions, our model disentangles the foreground object (modeled in 3D) from the background (modeled in 2D). In contrast to existing approaches, our method requires only a single forward pass to generate a full 3D scene. It hence allows for efficient rendering from arbitrary viewpoints while yielding 3D consistent results with high visual fidelity. Code and models are available at https://github.com/autonomousvision/voxgraf.

## 1 Introduction

Generating photorealistic renderings of scenes at high resolution is a long-standing goal in computer vision and graphics. The primary paradigm is to carefully design 3D models, which are then rendered using realistic camera and illumination models. In recent years, the computer vision community has made significant headway towards reducing these design efforts by approaching content generation from a data-centric perspective. Generative Adversarial Networks (GANs) [9] have emerged as a powerful class of generative models for photorealistic high-resolution image synthesis [2, 16, 17, 19, 20, 35, 36]. One benefit of these 2D models is that they can be trained with large collections of images which are readily available. However, scaling GANs to 3D is non-trivial because 3D supervision is difficult to obtain. Recently, *3D-aware* GANs have emerged to address the gap between handcrafted 3D models and image synthesis with 2D GANs which lack 3D constraints [4, 13, 23, 29, 30, 37]. 3D-aware GANs combine 3D generators, differentiable rendering and adversarial training to synthesize novel images with explicit control over the camera pose and, potentially, other scene properties like object shape and appearance.

Early 3D-aware GANs explored voxel-based 3D representations [13, 29]. To compensate for the cubic memory growth of voxel grids, HoloGAN [29] generates features on a small 3D grid and uses a neural network to map 3D features to a 2D image. While such a *neural renderer* allows scaling to higher image resolutions, it may also entangle viewpoint and generated content [37].

36th Conference on Neural Information Processing Systems (NeurIPS 2022).

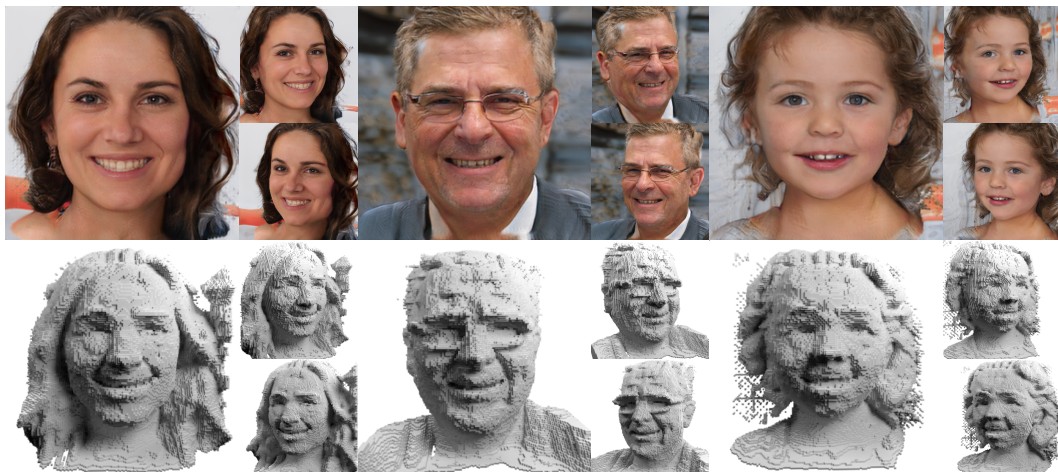

Figure 1: **3D-aware image synthesis with sparse voxel grids.** We investigate neural radiance fields represented as sparse voxel grids in the context of 3D-aware generative modeling. While training from unstructured image collections, our method allows for high quality image synthesis with explicit control over the camera viewpoint. In contrast to previous works, our method generates a 3D scene in a single forward pass allowing for more efficient and 3D-consistent rendering.

Consequently, early voxel-based approaches were either limited in image resolution or lacking 3D consistency. About the same time, Neural Radiance Fields (NeRF) [27] emerged in the context of view synthesis as a powerful alternative 3D representation. In their seminal work, Mildenhall et al. [27] represent a scene as a function of color and density, parameterized by a coordinate-based MLP. The predicted color and density values are then projected to an image with differentiable volume rendering. GRAF [38] adapts NeRF's coordinate-based representation to Generative Radiance Fields (GRAF) and proposes a 3D-aware GAN using a coordinate-based MLP and volume rendering. This propelled 3D-aware image synthesis to higher image resolutions while better preserving 3D consistency due to the physically-based and parameter-free rendering. These benefits led to the establishment of coordinate-based MLPs as new de facto standard for 3D-aware image synthesis [4, 31]. While recent 3D-aware GANs [4, 6, 10] have started to attain image fidelity and resolution similar to 2D GANs, training and inference is computationally expensive as the MLP must be queried at multiple points along each ray for volume rendering. However, querying 3D space densely is prohibitively costly. For example, rendering an image at resolution $256^2$ using $48$ sample points along each ray requires to query the neural network $256^2 \cdot 48 \approx 3M$ times. As a result, most recent works combine neural and volume rendering to ease computational cost at high resolutions [30]. Consequently, viewpoint and 3D content are often entangled, such that changing the camera pose might result in unwanted changes of the geometry or the appearance of the 3D scene. Further, for many downstream applications, e.g. integrating assets into physics engines, it is desirable to generate 3D content at high resolution directly. These shortcomings identify the need for a 3D representation that can be efficiently rendered at high resolution with a model that is *3D-consistent by design*.

Recently, there has been significant progress towards accelerated training of novel view synthesis models for single scenes by removing the MLP from the representation. In particular, DVGO [40] and Plenoxels [1] directly optimize a sparse voxel grid for novel-view synthesis, demonstrating that the visual fidelity attained by NeRF [27] is not primarily attributed to its MLP-based representation but rather to volumetric rendering and gradient-based optimization. In addition to impressive image quality, [1, 40] obtain large rendering speedups due to their fast density and color queries. Taking inspiration from these works, we revisit voxel-based representations for 3D-aware GANs [13, 29] in the context of volumetric rendering. To circumvent the cubic memory growth that limits early voxel-based approaches [13, 29], we explore *sparse* voxel grids for the generative settings, see Fig. 1. We observe that sparsity is key to enable scaling the 3D representation to higher resolution and to combine it with volume rendering. Specifically, we propose a 3D-aware GAN with a sparse voxel grid generator at its core. As a result, our approach inherits fast rendering and trilinear interpolation while being 3D-consistent by design, separating it from other recent 3D-aware GANs [3, 6, 10] which require a forward pass for every point along each camera ray of every view. Another difference to

existing 3D-aware GANs is that sparsity is a built-in feature of our representation, mitigating the need for exploiting sophisticated strategies to sample points along camera rays as in [4, 33, 43]. Our final model achieves image fidelity similar to recent 3D-aware GANs leveraging neural rendering while generating high-resolution geometry and improving 3D consistency. During inference, our model only requires a single forward pass which takes up most of the inference time. Once the 3D scene is generated, images can be rendered within milliseconds while existing approaches require another forward pass which is two orders of magnitude slower. We refer to our model as *VoxGRAF*.

## 2 Related Work

**2D GANs.** Rapid progress on Generative Adversarial Networks [9] now enables photorealistic synthesis up to megapixel resolution [2, 16, 18–20, 26, 36]. While the disentangled style-space of StyleGANs [18–20] allows for control over the viewpoint of the generated images to some extent [12, 24, 39, 48], gaining precise 3D-consistent control is still non-trivial due to its lack of physical interpretation and operation in 2D. In contrast, in this work we aim for explicit control over the camera pose by incorporating a 3D representation into the generator.

**3D-Aware GANs.** The first 3D-aware GANs, i.e. GANs that incorporate a 3D representation into the generator model, were voxel-based approaches. Dense grid-based approaches [13, 50] are limited to a lower grid resolution due to their cubic memory growth. Other works combine lower-resolution grids with neural rendering [23, 29] which scale to higher resolutions, but the generated images lack 3D consistency [38]. Recently, GRAF [37] and $\pi$-GAN combine volume rendering and coordinate-based representations [27] allowing to scale 3D-aware GANs with physically inspired rendering to high resolutions. However, dense ray marching remains computationally expensive and limits image fidelity. GIRAFFE [31] therefore proposes a hybrid rendering approach. They render a low-resolution feature map with ray marching and use a neural renderer to decode it into a high-resolution image. Due to efficiency, this approach has been widely adopted in subsequent works [3, 10, 15, 32, 44, 47, 49]. While some approaches try to counteract introduced inconsistencies, e.g. via dual discrimination [3] or a reconstruction loss [10], we instead propose a model that is *3D-consistent by design* and can generate the 3D object at high-resolution.

As an alternative to hybrid rendering, GOF [43], ShadeGAN [33] and GRAM [6] focus on reducing the number of query points for volume rendering. While aforementioned methods aim for reducing the number of sample points as querying a large MLP is computationally expensive, we instead use a sparse voxel grid as 3D representation. This allows us to speed up rendering without reducing the sample size as feature querying via trilinear interpolation is fast and can be efficiently implemented via custom CUDA kernels.

**Sparse 3D Representations.** As NeRF requires an optimization time in the order of multiple days per scene, a series of follow-up works [25, 34, 41, 46] propose techniques to speed up this process. The recent works Plenoxels [1] and DVGO [40] demonstrate that sparse voxel grid representations can achieve even faster convergence and higher rendering speed. In addition to efficient rendering, sparse voxel grids enable fast trilinear interpolation when queried beyond their grid resolution. Building on this representation, our approach inherits these benefits. We remark that very recently Instant-NGP [28] achieves even faster rendering by combining small MLPs with a multi-resolution hash table. Exploring this representation might be an interesting avenue for future extensions of our work. Note that all of these works focus on novel-view-synthesis for single scenes and require multi-view image supervision. Instead, we propose a generative model that trains with raw image collections and that can generate multiple novel instances at inference.

## 3 Method

We first provide the necessary background by summarizing the currently dominating paradigm for designing 3D-aware GANs which combines an MLP scene representation and volume rendering as introduced in GRAF [38]. Next, we introduce our sparse voxel-based scene representation which boosts rendering speed while retaining 3D-consistency by design. We refer to our model as VoxGRAF.

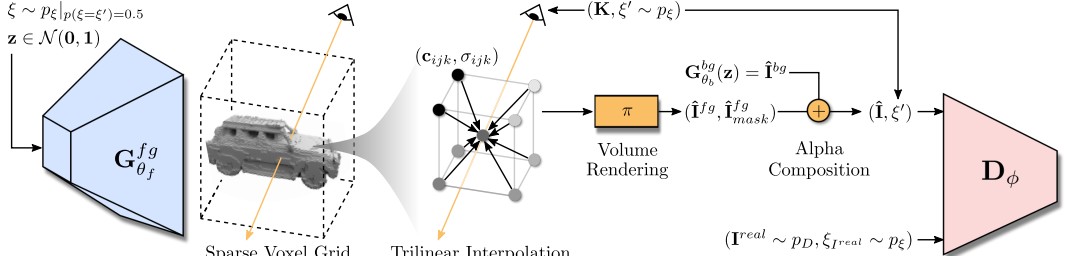

Figure 2: **VoxGRAF.** Conditioned on a camera pose $\xi$, the foreground generator $\mathbf{G}_{\theta_f}^{fg}$ maps a latent code $\mathbf{z}$ to color values $\mathbf{c} \in \mathbb{R}^{3 \times R_G \times R_G \times R_G}$ and densities $\sigma \in \mathbb{R}^{1 \times R_G \times R_G \times R_G}$ on a sparse voxel grid of resolution $R_G$. Given camera intrinsics $\mathbf{K}$ and a camera pose $\xi'$, a foreground image is obtained using differentiable volume rendering [27]. The values at the sampling points along the camera rays are computed by trilinearly interpolating the voxel grid [1]. The background is generated by a 2D GAN $\mathbf{G}_{\theta_b}^{bg}$ and combined with the foreground using alpha composition. The discriminator $D_\phi$ compares the generated image $\hat{\mathbf{I}}$ to the real image $\mathbf{I}^{real}$.

## 3.1 3D-aware GANs with Coordinate-Based Scene Representations

Recent 3D-aware GANs typically consist of a learned coordinate-based MLP as 3D generator, a deterministic volume rendering step potentially combined with a learned neural renderer in the 2D image domain, and a learned 2D discriminator. Let $G_\psi^{MLP}$ denote the 3D generator parameterized by a coordinate-based MLP with learnable parameters $\psi$. The 3D generator predicts a radiance field defined by color $\mathbf{c} \in [0,1]$ and density values $\sigma \in \mathbb{R}^+$. The radiance field is defined at any 3D point $\mathbf{x} \in \mathbb{R}^3$ and for any viewing direction $\mathbf{d} \in \mathbb{S}^2$. To model different 3D scenes, the generator is additionally conditioned on an $M$-dimensional latent variable $\mathbf{z} \in \mathcal{N}(\mathbf{0},\mathbf{1})$. The inputs $\mathbf{x}$ and $\mathbf{d}$ are projected to higher-dimensional features $\gamma(\mathbf{x}) \in L_\mathbf{x}$ and $\gamma(\mathbf{d}) \in L_\mathbf{d}$ with a fixed positional encoding to overcome the MLP's smoothness bias and model high-frequency content $\gamma(\cdot)$ [27,42]. Formally,

$$G_\psi^{MLP} : \mathbb{R}^{L_\mathbf{x}} \times \mathbb{R}^{L_\mathbf{d}} \times \mathbb{R}^M \to \mathbb{R}^3 \times \mathbb{R}^+ \qquad (\gamma(\mathbf{x}), \gamma(\mathbf{d}), \mathbf{z}) \mapsto (\mathbf{c}, \sigma) \qquad (1)$$

In this paper, we challenge this paradigm and investigate a sparse 3D CNN instead of a coordinate-based MLP as generator, as described in the next section.

The radiance field is rendered by approximating the intractable volumetric projection integral via numerical integration. First, the generator is queried at $N$ sampling points along each camera ray $r$ yielding colors and densities $\{(\mathbf{c}_r^i, \sigma_r^i)\}_{i=1}^N$. For each camera ray $r$, these points are projected to an RGB color value $\mathbf{c}_r$ and optionally an alpha mask $\mathbf{a}_r$ using alpha composition

$$\pi : (\mathbb{R}^3 \times \mathbb{R}^+)^N \to \mathbb{R}^3 \qquad \{(\mathbf{c}_r^i, \sigma_r^i)\} \mapsto \mathbf{c}_r$$

$$\mathbf{c}_r = \sum_{i=1}^N T_r^i \, \alpha_r^i \, \mathbf{c}_r^i \qquad \mathbf{a}_r = \sum_{i=1}^N T_r^i \, \alpha_r^i \qquad T_r^i = \prod_{j=1}^{i-1} \left(1 - \alpha_r^j\right) \qquad \alpha_r^i = 1 - \exp\left(-\sigma_r^i \delta_r^i\right) \quad (2)$$

where $T_r^i$ and $\alpha_r^i$ denote the transmittance and alpha value of sample point $i$ along ray $r$ and $\delta_r^i = \left\| \mathbf{x}_r^{i+1} - \mathbf{x}_r^i \right\|_2$ is the distance between neighboring sample points. As volume rendering has proven a powerful tool for high-fidelity reconstruction, VoxGRAF retains this rendering mechanism but reduces its computational cost by leveraging sparse scene representations.

## 3.2 VoxGRAF: Generating Radiance Fields on Sparse Voxel Grids

Our goal is to design a 3D-aware GAN based on a sparse scene representation that allows for efficient rendering. Fig. 2 shows an overview over our approach. In contrast to recent works [3,6,43], we do not use a coordinate-based MLP to parameterize the radiance field. Instead, inspired by recent work on novel view synthesis [1,40], we generate values on a sparse voxel grid using a 3D convolutional neural network. Therefore, our generator requires only a single forward pass to generate a 3D scene. To disentangle 3D content from the background we combine a 3D foreground generator $\mathbf{G}_{\theta_f}^{fg}$ with a 2D background generator $\mathbf{G}_{\theta_b}^{bg}$. $\mathbf{G}_{\theta_f}^{fg}$ takes a camera matrix $\mathbf{K}$, camera pose $\xi$ and a latent code $\mathbf{z}$

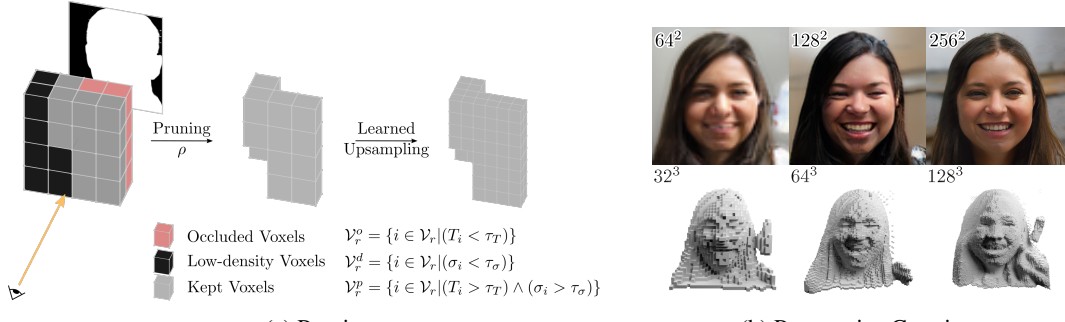

| | |
|---|---|
| Occluded Voxels | $\mathcal{V}_r^o = \{i \in \mathcal{V}_r | (T_i < \tau_T)\}$ |
| Low-density Voxels | $\mathcal{V}_r^d = \{i \in \mathcal{V}_r | (\sigma_i < \tau_\sigma)\}$ |
| Kept Voxels | $\mathcal{V}_r^p = \{i \in \mathcal{V}_r | (T_i > \tau_T) \wedge (\sigma_i > \tau_\sigma)\}$ |

(a) Pruning  (b) Progressive Growing

Figure 3: **Pruning and Progressive Growing.** A sparse representation is key for scaling voxel grids to high resolution. To sparsify the generated voxel grids, we combine density-based pruning (a) and progressive growing (b) while training our model. See text for details.

as input and predicts colors $\mathbf{c}$ and density values $\sigma$ on a sparse voxel grid. For unstructured images, camera matrix $\mathbf{K}$ and pose $\xi$ can be determined e.g. with an off-the-shelf pose detector as done in [4]. Volume rendering yields a foreground image and an alpha mask. The background generator maps the latent code $\mathbf{z}$ to a background image which is then combined with the foreground image using alpha composition. We train our model in an adversarial setting using a 2D discriminator on the full image.

**Foreground Generator.** Our foreground generator builds on the popular StyleGAN2 architecture [19] which achieves high fidelity in the 2D image domain. Conditioned on a camera pose $\xi \in \mathbb{R}^P$, it maps a latent code $\mathbf{z}$ to color values $\mathbf{c}$ and density values $\sigma$ on a sparse voxel grid

$$\mathbf{G}_{\theta_f}^{fg} : \quad \mathbb{R}^M \times \mathbb{R}^P \to \mathbb{R}^{3 \times R_G \times R_G \times R_G} \times \mathbb{R}^{1 \times R_G \times R_G \times R_G} \quad (\mathbf{z}, \xi) \mapsto (\mathbf{c}, \sigma) \tag{3}$$

where $\theta_f$ and $R_G$ denote the learnable parameters and the resolution of the voxel grid, respectively. Note that in contrast to most existing coordinate-based 3D-aware GANs, see Eq. (1), we do not condition the 3D generator on the view direction (per-ray). Instead, we follow [3] and condition it on the pose $\boldsymbol{\xi}$ (per-image) to model directional dependencies. For rendering, we compute $(\mathbf{c}_r^i, \sigma_r^i)$ via trilinear interpolation of densities and colors stored at the nearest eight vertices [1]. We use the same rendering formulation outlined in 3.1 and leverage custom CUDA kernels[1] for efficiency.

To generate voxel grids instead of images, we replace all 2D operations of the StyleGAN2 generator with their 3D equivalent, e.g., 3D modulated convolutions and 3D upsampling. At resolutions beyond $32^3$, we investigate sparse convolutions [11] instead of dense ones as they can be more computationally efficient. We compare the computational efficiency of sparse convolutions[2] to dense convolutions for which we zero out the values of pruned voxels. For our architecture, using sparse convolutions reduces the memory consumption but due to the computational overhead for managing coordinates increases the runtime, see Table 1. To sparsify the representation, we combine progressive growing [16] with pruning [1] as illustrated in Fig. 3a. Specifically, we start with training a dense model at resolution $32^3$. After sufficient training, we add the next layer of convolutions and prune its inputs based on the rendered view. Intuitively, the next layer should only operate on voxels visible in the rendered view to yield a sparse representation. Consequently, we prune voxels that are either occluded or have a low density. Following Eq. (2), the rendered alpha value is $\mathbf{a}_r = \sum_{i=1}^N T_r^i \alpha_r^i$. Accordingly, occluded voxels (low transmittance $T_i$) and empty voxels (low density $\sigma_i$) do not contribute to the final image. The pruning operator $\rho$ discards all voxels along a camera ray $r$ for which transmittance $T^i$ or density $\sigma_i$ is smaller than threshold $\tau_T$ or $\tau_\sigma$, respectively. Let $\mathcal{V}_r$ denote the set of voxels intersecting with ray $\mathbf{r}$, then

$$\rho : \mathcal{V}_r \mapsto \mathcal{V}_r^p \qquad \mathcal{V}_r^p = \{i \in \mathcal{V}_r | (T_i > \tau_T) \wedge (\sigma_i > \tau_\sigma)\} \tag{4}$$

where $\mathcal{V}_r^p$ is the set of the retained voxels. After pruning, we upsample the kept voxels via sparse transposed convolutions. After the newly-added layer is sufficiently trained, we repeat this process for the next added stage. Fig. 3b shows images and voxel grids at different resolutions. We implement this on-the-fly pruning operation using efficient custom CUDA kernels.

---

[1] We build on the kernels from https://github.com/sxyu/svox2.git
[2] We use the Minkowski Engine library https://github.com/NVIDIA/MinkowskiEngine.git

In agreement with [3], we observe that it is crucial to account for view dependent effects or pose-correlated attributes in the training data, like eyes always looking into the camera. We take two measures to increase the flexibility of our model: Following [3], $\mathbf{G}_{\theta_f}^{fg}$ is conditioned on the pose $\xi$ which corresponds to the rendering pose $\xi'$ in $50\%$ of the cases and is randomly chosen otherwise. Formally, $\xi \sim p_\xi|_{p(\xi=\xi')=0.5}$. At inference, the pose-conditioning is fixed to retain 3D consistency. However, we find that pose conditioning alone is not always sufficient to account for strong correlations in the data. Depending on the dataset, we optionally refine the rendered image with a shallow 2D CNN with 2 hidden layers of dimension 16 and kernel size 3. While this refinement is powerful enough to model dataset biases, it is considerably less flexible than the neural rendering used in [4, 10, 30]: Our 2D CNN operates on the rendered image instead of rendered features at smaller resolution. Operating on 3 channels at full resolution allows to keep its capacity at a minimum and, due to not using any upsampling operation, results in a local receptive field.

**Background Generator.** We consider datasets with a single object per image. Modeling only the object in 3D saves computation and is advantageous for potential downstream tasks, e.g., integrating generated assets into new environments Hence, we model the object in 3D and generate the background of the image with a 2D GAN. Specifically, we use the StyleGAN2 generator [19] with reduced channel size as modeling the background requires less capacity than generating the full image:

$$\mathbf{G}_{\theta_b}^{bg}: \quad \mathbb{R}^M \to \mathbb{R}^{3 \times R_I \times R_I} \quad \mathbf{z} \mapsto \hat{\mathbf{I}}^{bg} \tag{5}$$

We choose the same latent code $\mathbf{z}$ for foreground and background to allow for modeling correlations like lighting between the two. Note that, unlike the foreground generator, the background generator is not conditioned on the camera pose. As the background remains fixed when changing the camera pose, the generator is encouraged to model pose-dependent content with the foreground generator leading to disentanglement (see supp. mat. for qualitative disentanglement results).

The final image is obtained using alpha composition

$$\hat{\mathbf{I}} = \hat{\mathbf{I}}_{mask}^{fg} \cdot \hat{\mathbf{I}}^{fg} + (1 - \hat{\mathbf{I}}_{mask}^{fg}) \cdot \hat{\mathbf{I}}^{bg}$$

The full generator is defined as

$$\mathbf{G}_\theta: \quad \mathbb{R}^M \times \mathbb{R}^P \times \mathbb{R}^P \times \mathbb{R}^K \to \mathbb{R}^{3 \times R_I \times R_I} \quad (\mathbf{z}, \xi, \xi', \mathbf{K}) \mapsto \hat{\mathbf{I}} \tag{6}$$

**Regularization.** For fast rendering, it is crucial that most generated voxels are either fully opaque or empty such that early stopping and empty space skipping are effective. By regularizing the variance of the expected depth $\hat{z}_r$ along each ray $r$, the foreground generator is encouraged to generate a single, sharp surface:

$$\hat{z} = \sum_i T_i \alpha_i z_i \qquad Var(\hat{z}) = \frac{1}{\sum_i T_i \alpha_i} \sum_j T_j \alpha_j (z_j - \hat{z})^2 \tag{7}$$

$$\mathcal{L}_{DV} = \lambda_{DV} \max(Var(\hat{z}), \tau) \tag{8}$$

where $\tau$ is a hyperparameter that defines the thickness of the surface. We find that thresholding the loss is important to avoid an empty foreground. In addition, we find that adding the grid TV regularization $\mathcal{L}_{TV}$ from [1] and fore- and background coverage regularization $\mathcal{L}_{cvg}^{fg}$ and $\mathcal{L}_{cvg}^{bg}$ from [45] further stabilizes training (see sup. mat. for details). The full regularization term of our generator is

$$\mathcal{L}_{\text{reg}} = \mathcal{L}_{DV} + \mathcal{L}_{TV} + \mathcal{L}_{cvg}^{fg} + \mathcal{L}_{cvg}^{bg} \tag{9}$$

**Discriminator.** We use the StyleGAN2 discriminator and condition it on the camera pose as proposed in [3]. Similarly, we find that conditioning guides the generator to learn correct 3D priors and a canonical representation. Since rendering our sparse representation is fast, our discriminator is able to operate on the full image and does not need to consider image patches as done in GRAF [38].

### 3.3 Training

Given images $\mathbf{I}$ from the data distribution $p_\mathcal{D}$ with known camera extrinsics $\boldsymbol{\xi}_\mathbf{I}$ and intrinsics $\mathbf{K}_\mathbf{I}$ and latent codes $\mathbf{z} \in \mathcal{N}(\mathbf{0}, \mathbf{1})$, we train our model using a GAN objective with R1-regularization [26]

$$V(\theta, \phi) = \mathbb{E}_{\mathbf{z} \sim \mathcal{N}(\mathbf{0},\mathbf{1}), \xi, \xi' \sim p_\xi} \left[ f(-D_\phi (G_\theta(\mathbf{z}, \boldsymbol{\xi}, \boldsymbol{\xi}', \mathbf{K}), \boldsymbol{\xi}')) \right] \tag{10}$$

$$+ \mathbb{E}_{\mathbf{I} \sim p_\mathcal{D}} \left[ f(D_\phi(\mathbf{I}, \boldsymbol{\xi}_\mathbf{I})) - \lambda \|\nabla D_\phi(\mathbf{I}, \boldsymbol{\xi}_\mathbf{I})\|^2 \right] \tag{11}$$

where $f(t) = -\log(1 + \exp(-t))$ and $\lambda$ controls the strength of the R1-regularizer. $G_\theta$ and $D_\phi$ are trained with alternating gradient descent combining the GAN objective with the regularization terms:

$$\min_\theta \max_\phi \quad V(\theta, \phi) + \mathcal{L}_{reg}(\theta) \tag{12}$$

In practice, we optimize the generator with a non-saturating variant of Eq. (12) [9]. We train our approach with Adam [21] using a batch size of 64 at grid resolution $R_G = 32, 64$ and 32 at $R_G = 128$. We use a learning rate of $0.0025$ for the generator and $0.002$ for the discriminator. For faster training, we first grow $R_I$ from 32 to 128 while keeping $R_G$ at 32. Then, we alternately increase the grid resolution and the image resolution until the dataset resolution and $R_G = 128$ are reached. For synthetic datasets, i.e. Carla [38], we do not add any refinement layers. Depending on the dataset, we train our models for 3 to 7 days on 8 Tesla V100 GPUs. Details on the network architectures can be found in the supplemental material.

## 4 Results

**Datasets.** We validate our approach on standard benchmark datasets for 3D-aware image synthesis. The synthetic Carla dataset [8, 37] contains 10k images and camera poses of 18 car models with randomly sampled colors. FFHQ [19] comprises 70k aligned face images. AFHQv2 Cats [5] consists of 4834 cat faces. Following [4], we estimate camera poses for both datasets with off-the-shelf pose estimators [7, 22] and augment all datasets with horizontal flips. Due to the limited number of images in AFHQv2 and Carla, we use adaptive discriminator augmentation [17] for these datasets.

**Evaluation Metrics.** We measure image fidelity by calculating the Fréchet Inception Distance (*FID*) [14] between 20k generated images and the full dataset. For all runtime comparisons, we report times on a single Tesla V100 GPU with a batch size of 1.

### 4.1 Ablation Study

We investigate the sparsity of the generated voxel grids and validate the importance of the depth variance loss, see Eq. (8). Sparsity is evaluated by fusing the pruned voxel grids from 16 equally spaced camera views and reporting the number of empty voxels divided by the total number of voxels $R_G^3$. Table 1 shows results averaged over 100 instances. The depth variance loss increases sparsity from 74% to 95%, lowering the memory consumption. With the sparser representation, the times needed for scene generation $t_{G f g + b g}$ and rendering $t_\pi$ are reduced significantly. We

|  | Sparsity [%] | Memory [GB] | $t_{Gfg+bg}$ [ms] | $t_\pi$ [ms] |
|---|---|---|---|---|
| w/o $\mathcal{L}_{\mathcal{DV}}$ | 74 | 1.4 | 229 | 7 |
| w $\mathcal{L}_{\mathcal{DV}}$ | 95 | 0.9 | 198 | 4 |
| w $\mathcal{L}_{\mathcal{DV}}{}^\dagger$ | 95 | 1.1 | 58 | 4 |

Table 1: **Regularization.** We compare sparsity and rendering time with and without depth variance regularization $\mathcal{L}_{\mathcal{DV}}$ on FFHQ with $R_G = 128$ and $R_I = 128$. $^\dagger$ denotes an architecture with dense convolutions where values of pruned voxels are set to zero.

further compare an implementation with dense instead of sparse convolutions where we zero out the values of pruned voxels. While this increases memory consumption, it reduces the runtime due to the computational overhead of managing coordinates for sparse convolutions. We prioritize faster training over memory and hence train our models with dense convolutions where we set values of pruned voxels to zero.

### 4.2 Baseline Comparison

**Baselines.** We group the baselines into two categories: (i) methods that render low-resolution features and use 2D-upsampling, i.e. a neural renderer, to obtain the final image, e.g. StyleNeRF [10], and (ii) methods that render the 3D representation directly at the final image resolution, e.g. $\pi$-GAN [4]. VoxGRAF falls into the second group. We use the official code release for all methods. Further, we reference numbers reported by concurrent works GRAM [6] and EG3D [3].

**Qualitative Results.** Fig. 4 shows samples from multiple views for StyleNeRF, GRAM and VoxGRAF on FFHQ at resolution $256^2$. StyleNeRF can generate additional faces or strands of hair across views, as indicated by the red frames in Fig. 4. These inconsistencies under viewpoint changes

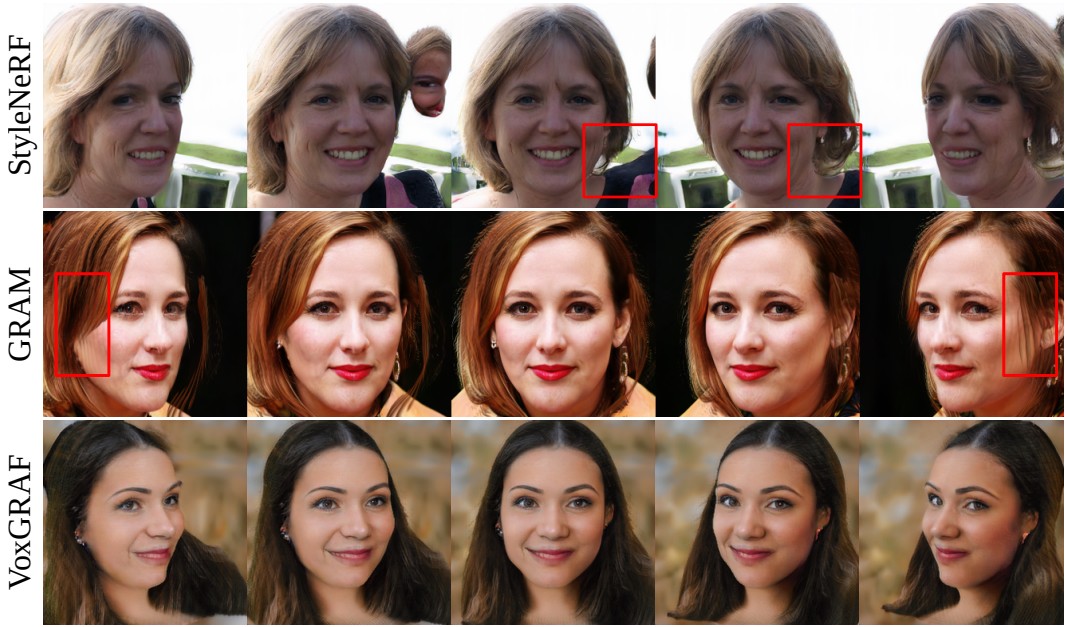

Figure 4: **Qualitative Comparison on Multi-View Consistency**. We compare multi-view consistency of state-of-the-art baselines and our method. While StyleNeRF's powerful neural renderer can introduce multi-view inconsistencies (appearing faces, moving strands of hair), GRAM's manifold representation becomes visible in layered artifacts for steeper viewing angles. In contrast, our method leads to more multi-view consistent results.

|  | FFHQ [19] | AFHQ [5] | Carla [38] |
|---|---|---|---|
|  | $R_I = 256^2$ | $R_I = 256^2$ | $R_I = 128^2$ |
| GIRAFFE [31] | 31.5 | 16.1 | – |
| VolumeGAN [44] | 9.1 | – | 7.9 |
| StyleNeRF [10] | 8.0 | – | – |
| EG3D [3] | 4.8 | 3.9 | – |
| GRAF [38] | 71 | 121 | 41 |
| $\pi$-GAN [4] | 85 | 47 | 29.2 |
| GOF [43] | 69.2 | 54.1 | 29.3 |
| GRAM [6] | 17.9 | 18.5 | 26.3 |
| VoxGRAF | 9.6 | 9.6 | 6.7 |

Table 2: **Quantitative Comparison.** We compare against state-of-the-art methods with a neural rendering pipeline (first block) and without one (second block). We report FID [14] between 20k generated images and the full dataset.

|  | $R_I = 128^2$ | $R_I = 256^2$ |
|---|---|---|
| GIRAFFE [30] | – | 5 |
| StyleNeRF [10] | – | 49 |
| EG3D* [3] | – | 27 |
| GRAF [38] | 219 | 878 |
| $\pi$-GAN [4] | 154 | 608 |
| GOF [43] | 199 | 742 |
| GRAM [6] | 136 | 418 |
| VoxGRAF | 58 + 3 | 58 + 6 |

Table 3: **Rendering times**. We report time in ms per image. Note that our method allows for separating scene generation (first number) and rendering (second number) which is useful for real-time rendering applications. *EG3D is evaluated on a faster GPU (RTX 3090 GPU) compared to the others (Tesla V100 GPU).

are introduced by StyleNeRF's powerful neural renderer. GRAM achieves more consistent results, but its plane representation creates stripe artifacts for large viewpoint ranges. Due to the shallow 2D CNN, VoxGRAF can model the dataset bias of eyes looking into the camera but otherwise achieves high 3D-consistency even under large viewing angles. Additional samples of our method and the corresponding sparse voxel grids for all datasets are provided in Fig. 1 and Fig. 5.

**Quantitative Results.** Table 2 reports FID on all datasets. As expected, methods that use neural rendering with upsampling, i.e., StyleNeRF and EG3D, perform best in terms of image fidelity. This is expected as a neural renderer can add flexibility. But, as shown in Fig. 4, for StyleNeRF it reduces 3D-consistency. Among methods without a neural renderer, VoxGRAF significantly improves over $\pi$-GAN and GOF and surpasses the current state-of-the-art approach GRAM.

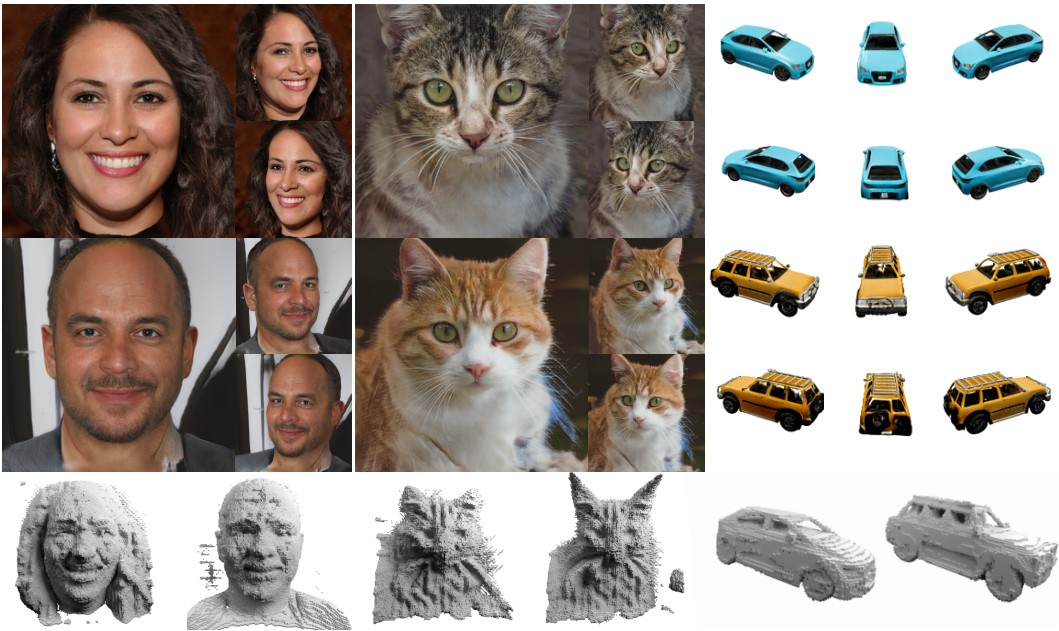

Figure 5: **Qualitative Results for our Method.** We show generated images at resolution $256^2$ for FFHQ [19] and AFHQ [5] and samples at resolution $128^2$ for Carla [38].

**Runtime Comparison.** Lastly, we compare the rendering times at inference for methods with and methods without a neural renderer. In contrast to all baselines, VoxGRAF requires only a single forward pass to generate the scene, which can then be rendered from different viewpoints efficiently. Note that at inference, voxels are pruned solely based on their density to amortize the rendering costs per scene. We find that this does not visibly affect the rendered images. We report the times for generating the scene and rendering one view separately in Table 3. One of the earliest works, GIRAFFE, is the fastest among all approaches as it renders a low-resolution feature volume and uses comparably small neural networks. StyleNeRF significantly increases the neural renderer's size to improve image fidelity, which comes at the cost of speed compared to GIRAFFE. Yet, StyleNeRF is the fastest approach for generating a single image among the best-performing methods. However, a potential application of 3D-aware GANs is generating novel views of a single instance in real-time. In this setting, at resolution $256^2$, VoxGRAF generates novel views at 167 FPS, whereas StyleNeRF runs at 20 FPS as its rendering costs are not amortized per scene.

## 5 Limitations and Discussion

In this work, we investigate sparse voxel grids as representation for 3D-aware image synthesis. We find that the key to generating sparse voxel grids is to combine progressive growing, pruning, and regularization to encourage a sharp surface that can be rendered efficiently. Our approach outperforms all methods that do not employ a neural renderer. Instead of discarding neural rendering entirely, we find it advantageous to utilize a shallow CNN for refinement. This CNN can model dataset bias but is significantly weaker than standard neural rendering approaches that upsample low resolution feature maps. Our approach can reduce the gap to models that build heavily on neural rendering, yet a trade-off between 3D-consistency and image fidelity remains. Whether a certain amount of neural rendering is inherently needed to reach best performance is an important direction for future research. Lastly, the speed of our method depends on the sparsity of the modeled scene. Therefore, rendering times will likely increase on more complex datasets than those commonly used in literature.

## Acknowledgments and Disclosure of Funding

We acknowledge the financial support by the BMWi in the project KI Delta Learning (project number 19A19013O), the support from the BMBF through the Tuebingen AI Center (FKZ:01IS18039A), and the support of the DFG under Germany's Excellence Strategy (EXC number 2064/1 - Project number 390727645). Andreas Geiger and Michael Niemeyer were supported by the ERC Starting Grant LEGO-3D (850533). We thank the International Max Planck Research School for Intelligent Systems (IMPRS-IS) for supporting Katja Schwarz and Michael Niemeyer. This work was supported by an NVIDIA research gift. We thank Christian Reiser for the helpful discussions and suggestions. Lastly, we would like to thank Nicolas Guenther for his general support.

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
