# Supplementary Material for
# VoxGRAF: Fast 3D-Aware Image Synthesis with Sparse Voxel Grids

**Katja Schwarz**[1]    **Axel Sauer**[1]    **Michael Niemeyer**[1]    **Yiyi Liao**[2]    **Andreas Geiger**[1]

[1]University of Tübingen and Max Planck Institute for Intelligent Systems, Tübingen
[2] Zhejiang University, China

## Abstract

In this **supplementary document**, we first provide details on the network architectures and training strategy of our approach in Section 1. Our settings for the baselines are described in Section 2. Section 3 shows additional results and failure cases. In Section 4, we discuss the societal impact of our work. The **supplementary video** includes synthesized animations in which we control the camera viewpoint for our method and the baselines and show interpolations between latent codes. We use the same mathematical notation as in the paper.

## 1  Implementation

**Foreground Generator**  The foreground generator builds on the StyleGAN2 generator [14] replacing 2D operations with their 3D equivalent as described in the main paper. For faster training, we consider the layers of StyleGAN2 instead of their alias-free version proposed in StyleGAN3 [13]. The mapping network has 2 layers with 64 channels. Since 3D convolutions have more parameters than 2D convolutions with the same channel size, we reduce the channel base[1] from 32768 for StyleGAN2 to 4000. To facilitate progressive growing we choose an architecture with skip connections which adds an upsampled version of the output grid of the previous layer to the input of the next layer. The skip architecture is equivalent to the 2D variant proposed in [14].

Following [3], we condition the generator on a camera pose. Specifically, we condition the generator on a rotation matrix and a translation vector, yielding a 12-dimensional vector as input to the conditioning.

The foreground generator predicts color and density values on a sparse voxel grid. Following Plenoxels [1], the generator outputs the coefficients of spherical harmonics. We choose spherical harmonics of degree 0, i.e., a single coefficient for each color channel. For a sharp surface and efficient rendering, the foreground generator needs to predict high values for the density. We facilitate generating high values by multiplying the density output of the network with a factor of 30. A similar idea was proposed in [1] where the learning rate for the density is set to a higher value than the learning rate for the color.

**Sparse Convolutions**  We investigate sparse convolutions [8] for the foreground generator but find that the computational overhead of managing coordinates increases runtime for our architecture. We therefore use dense convolutions and zero out values in the feature maps for pruned voxels. We also compare the difference for both implementations on performance. In general, we observe similar training behavior for both implementations. However, the faster dense implementation allows us to

---

[1]see https://github.com/NVlabs/stylegan3.git

train the model for 60M iterations compared to 30M for the model with sparse convolutions. This improves FID from 14.4 for the sparse implementation to 9.0 for the dense implementation on FFHQ 256.

**Background Generator**  We use the StyleGAN2 [14] generator with a 2-layer mapping network with 64 channels, and a synthesis network with channel base 2048 and a maximum of 64 channels per layer.

**2D Refinement Layers**  Depending on the dataset, we optionally refine the rendered image with a shallow 2D CNN with 2 hidden layers of dimension 16 and kernel size 3. To avoid texture sticking under viewpoint changes we use alias-free layers [13] with critical sampling.

**Regularization**  Without regularization, volume rendering is prone to result in semi-opaque voxels and floating artifacts and struggles to accurately represent sharp surfaces [10, 16, 18, 21]. Therefore, we regularize both the variance of the depth, as described in the main paper, and the total variation of the predicted density. For the depth variance loss $\mathcal{L}_{DV}$, we set $\tau = (1.5\delta_0)^2$ where $\delta_0$ is the size of one voxel and $\lambda_{DV} = 0.01$.

Following Plenoxels [1], we regularize the total variation of the predicted density values in the set of all voxels $\mathcal{V}$ for a compact, smooth geometry

$$\mathcal{L}_{TV} = \lambda_{TV} \frac{1}{|\mathcal{V}|} \sum_{\mathbf{v} \in \mathcal{V}} \sqrt{\Delta_x^2(\sigma) + \Delta_y^2(\sigma) + \Delta_z^2(\sigma)} \tag{1}$$

with $\Delta_x^2(\sigma)$ shorthand for $(\sigma_{i,j,k} - \sigma_{i+1,j,k})^2$ and analogously for $\Delta_y^2(\sigma)$ and $\Delta_z^2(\sigma)$. For efficiency, we evaluate the loss stochastically on random contiguous segments of voxels as proposed in [1] and set $\lambda_{TV} = 10^{-5}$ in all experiments.

To avoid that the full image is generated by either background or foreground generator, we use a hinge loss on the mean mask value as proposed in GIRAFFE-HD [23]

$$\mathcal{L}_{cvg}^{fg} = \lambda_{cvg}^{fg} \max\left(0, \eta^{fg} - \frac{1}{|\mathcal{S}|} \sum_{i \in \mathcal{S}} \hat{\mathbf{I}}_{mask}^{fg}[i]\right) \tag{2}$$

$$\mathcal{L}_{cvg}^{bg} = \lambda_{cvg}^{bg} \max\left(0, \eta^{bg} - \frac{1}{|\mathcal{S}|} \sum_{i \in \mathcal{S}} 1 - \hat{\mathbf{I}}_{mask}^{fg}[i]\right) \tag{3}$$

where $\eta^{fg}$ and $\eta^{bg}$ denote the minimum fraction that should be covered by foreground and background, respectively. To ensure that both models are used, we set $\eta^{fg} = 0.4$ for AFHQ [5] and FFHQ [14] and $\eta^{fg} = 0.1$ for Carla [19] because Carla's objects cover a much smaller fraction of the image. We use $\eta^{bg} = 0.1$ and $\lambda_{cvg}^{fg} = \lambda_{cvg}^{bg} = 0.1$ for all datasets.

**Discriminator**  We use the StyleGAN2 [14] discriminator with conditional input as in [3]. To facilitate progressive growing we choose a skip architecture which adds a downsampled version of the input image to the input of each layer as introduced in [14].

**Rendering**  For efficient rendering, we leverage custom CUDA kernels building on the official code release of [1]. We select equidistant sampling points for volume rendering in steps of 0.5 voxels but skip voxels with $\sigma_i < 10^{-10}$ and stop rendering early if $T_i < 10^{-7}$ as in [1].

**Implementation and Training**  Our code base builds on the official PyTorch implementation of StyleGAN2 [14] available at https://github.com/NVlabs/stylegan3. Similar to StyleGAN2, we train with equalized learning rates for the trainable parameters and a minibatch standard deviation layer at the end of the discriminator [11] and apply an exponential moving average of the generator weights. For faster training, we use mixed-precision for both the generator and the discriminator as proposed in [12]. Unlike [14], we do not train with path regularization or style-mixing. To reduce computational cost and overall memory usage R1-regularization [15] is applied only once every 4 minibatches. We use a regularization strength of $\gamma = 1$ for all datasets. Due to the small size of AFHQ, we follow [12] and finetune a generator that is pretrained on FFHQ with $R_I = 128$ and $R_G = 32$, i.e., before the representation is pruned.

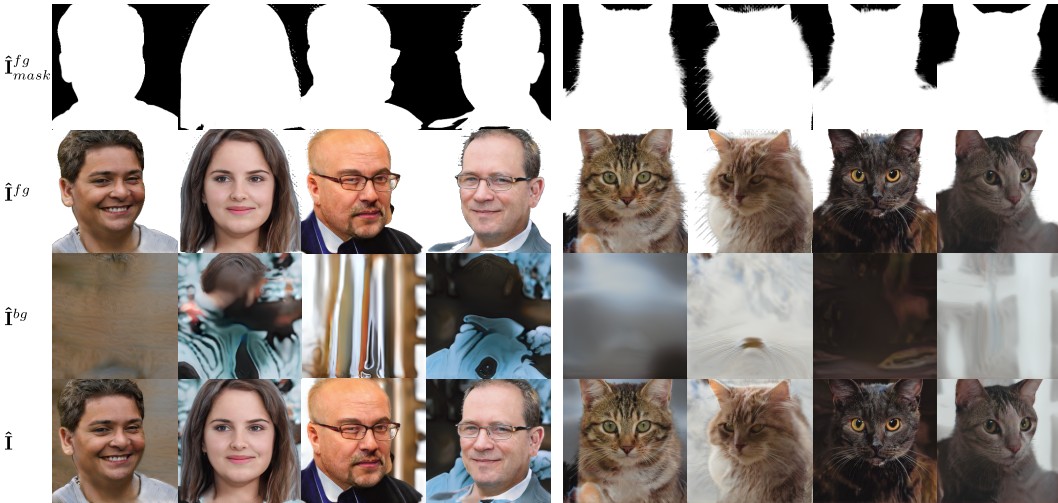

$\hat{\mathbf{I}}_{mask}^{fg}$

$\hat{\mathbf{I}}^{fg}$

$\hat{\mathbf{I}}^{bg}$

$\hat{\mathbf{I}}$

Figure 1: **Background disentanglement.** We show generated images at resolution $256^2$ for FFHQ [14] and AFHQ [5] with truncation $\psi = 0.7$.

## 2 Baselines

**Qualitative Results** We provide qualitative results for StyleNeRF [7] and GRAM [6] in both the main paper and the supplemental material. For StyleNeRF, we obtain samples from the pretrained FFHQ model available at https://github.com/facebookresearch/StyleNeRF.git. For GRAM, its authors kindly provided unpublished code and pretrained models which we use for evaluation. In the qualitative comparisons, i.e., Fig. 4 of the main paper and Fig. 3, we use a truncation of $\psi = 0.7$ for all methods. For GRAM and our approach, we show samples from $-40°$ to $+40°$ which roughly corresponds to 2 standard deviations of the pose distribution. We find that StyleNeRF does not necessarily adhere to the input pose. Hence, we manually define the range to be $-60°$ to $+60°$ such that the rendered images roughly align with the other methods.

**Quantitative Results** Table 2 of the main paper shows a quantitative comparison for all baselines and our method in terms of FID. For EG3D [3] and GIRAFFE [16], we report the numbers from [3]. For StyleNeRF [7], we take the numbers from [7]. From [7] we further reference the results on FFHQ and AFHQ for GRAF [20] and $\pi$-GAN [4] as these datasets were not considered in the original publications. On Carla, we report the results from [20] and [4], respectively. For GOF [21], we reference the numbers from [21] on Carla and train new models to obtain results on FFHQ and AFHQ using the official code release available at https://github.com/SheldonTsui/GOF_NeurIPS2021.git. For GRAM [6], we report the results from [6] for FFHQ. As AFHQ is not considered in [6], we train GRAM on AFHQ using their unpublished code. Similar to our approach, we finetune a generator that was pre-trained on FFHQ. We remark that across all reported values for FID the number of generated image varies where most methods report values considering either 20k or 50k generated images. An overview is provided in Table 1 which is discussed in more detail at the end of Section 3.

## 3 Results

**Background and Foreground Disentanglement.** Fig. 1 illustrates foreground masks, foreground and background image, and the image after alpha composition. As the background remains fixed under viewpoint changes, the generator is encouraged to model pose-dependent content with the foreground generator. The regularization in Eq. (2) and Eq. (3) encourages the generator to use both the background and the foreground generator to synthesize the full image.

**Multi-View Consistency.** Corresponding to Fig. 4 of the main paper, we provide additional qualitative comparisons on multi-view consistency in Fig. 3. For StyleNeRF [7], the red boxes highlight inconsistencies, like changing eye shape (first row on the left), moving strands of hair

| | FFHQ [14] $R_I = 256^2$ | AFHQ [5] $R_I = 256^2$ | Carla [19] $R_I = 128^2$ |
|---|---|---|---|
| GIRAFFE [17] | $31.5^\dagger$ | $16.1^\dagger$ | – |
| VolumeGAN [22] | $9.1^\dagger$ | – | $7.9^\dagger$ |
| StyleNeRF [7] | $8.0^\dagger$ | – | – |
| EG3D [3] | $4.8^\dagger$ | $3.9^\dagger$ | – |
| GRAF [19] | $71^\dagger$ | $121^\dagger$ | $41^{*1}$ |
| $\pi$-GAN [4] | $85^\dagger$ | $47^\dagger$ | $29.2^{*2}$ |
| GOF [21] | 69.2 | 54.1 | 29.3 |
| GRAM [6] | 17.9 | 18.5 | 26.3 |
| VoxGRAF | $9.6 / 9.0^\dagger$ | $9.6 / 9.4^\dagger$ | $6.7 / 6.3^\dagger$ |

Table 1: **Quantitative Comparison.** We report FID [9] on the full dataset and explicitly annotate the number of generated images for evaluation. $^\dagger$ denotes 50k generated images, $*1$ denotes 1k images (GRAF, Carla) and $*2$ denotes 8k images ($\pi$-GAN, Carla). Numbers without annotation are calculated using 20k generated images.

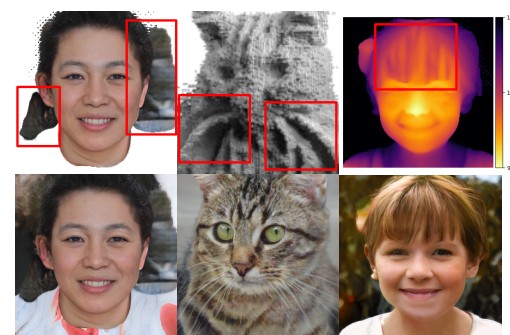

Figure 2: **Failure Cases.** Left: Some part of the background is modeled with the foreground. Middle: The whiskers are connected to the body of the cat. Right: The hair is directed inward to the head.

| | $\mathcal{L}_{reg}$ | w/o $\mathcal{L}_{DV}$ | w/o $\mathcal{L}_{TV}$ | w/o $\mathcal{L}_{cvg}^{fg}$ | w/o $\mathcal{L}_{cvg}^{bg}$ |
|---|---|---|---|---|---|
| FID | 14.2 | 14.8 | 15.1 | 14.6 | 14.8 |

Table 2: **Impact of Regularization on FID.** We ablate the performance of all four regularizers on models trained on FFHQ with $R_I = 128$ and $R_G = 64$. While the regularizers do not significantly impact end-to-end performance measured in FID, we find that they can be helpful to stabilize training.

| | Depth ↓ | Pose ↓ |
|---|---|---|
| StyleNeRF [7] | – | $0.051 \pm 0.047$ |
| GRAM [6] | $0.48 \pm 0.24$ | $0.013 \pm 0.013$ |
| EG3D [3] | $0.29 \pm 0.30$ | $0.0018 \pm 0.0031$ |
| VoxGRAF | $0.33 \pm 0.23$ | $0.00045 \pm 0.00079$ |

Table 3: **View-Consistency**. We re-implement the depth and pose metric from [3]. While results agree with the qualitative evaluation in Fig. 4 of the main paper, the large standard deviations indicate that both metrics are very sensitive to the latent code and the sampled poses. Note that for StyleNeRF depth can only be rendered at resolution $32^2$ and we thus omit evaluating the Depth metric for it.

(second row, third row on the left) and distortion of the face shape (first and third row on the right). For GRAM [6], red boxes indicate layered artifacts stemming from its manifold representation. In contrast, our method leads to more multi-view consistent results.

We further include a quantitative evaluation on consistency in Table 3. We implemented our own version of the depth and pose metric following the description in [3] as their evaluation code is not publicly available. We report the results for our approach, GRAM, StyleNeRF and EG3D for reference. We also report the standard deviation across the 1024 samples used for evaluation. While the results in Table 3 agree with our qualitative analysis of view consistency in Fig. 4 and the supplementary video, we find that both metrics are very sensitive to the latent code and the sampled poses, as indicated by the large standard deviations. As no established evaluation pipeline exists, our results should not be directly compared to the numbers in [3] as the implementation and pose sampling might differ.

**Pose Conditioning.** Fig. 4 illustrates the impact of the pose conditioning on the generated images. Pose conditioning is not only used to model slight changes, e.g. of the eyes or the smile, but can alter the general appearance of the generated instance. However, by fixing the pose conditioning during inference, view-consistent images can be generated.

**Regularization.** We ablate the effect of regularization in terms of end-to-end performance measured in FID. Table 2 shows that the regularizers do not significantly change FID. Nonetheless, $\mathcal{L}_{DV}$ speeds up training (see Table 1 of the main paper) and we find the remaining losses to be helpful for stabilizing training.

**Failure Cases.** Fig. 2 illustrates failure cases of out method. For some samples, we observe that the background and the foreground are not disentangled properly. The left column in Fig. 2 shows an example where the foreground generates parts of the background, as indicated by the red boxes. In

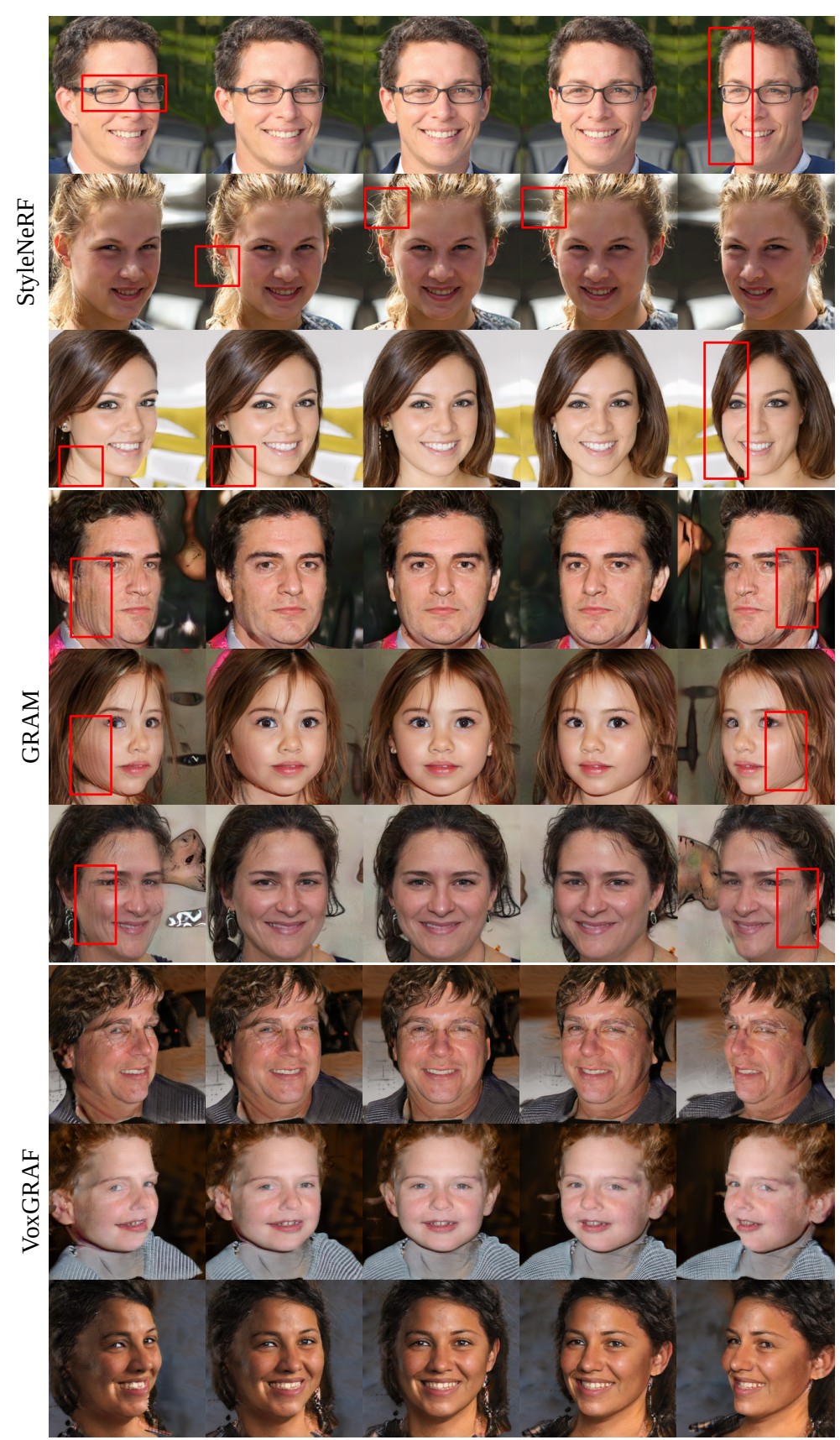

Figure 3: **Qualitative Comparison on Multi-View Consistency**. We apply truncation with $\psi = 0.7$ for all methods.

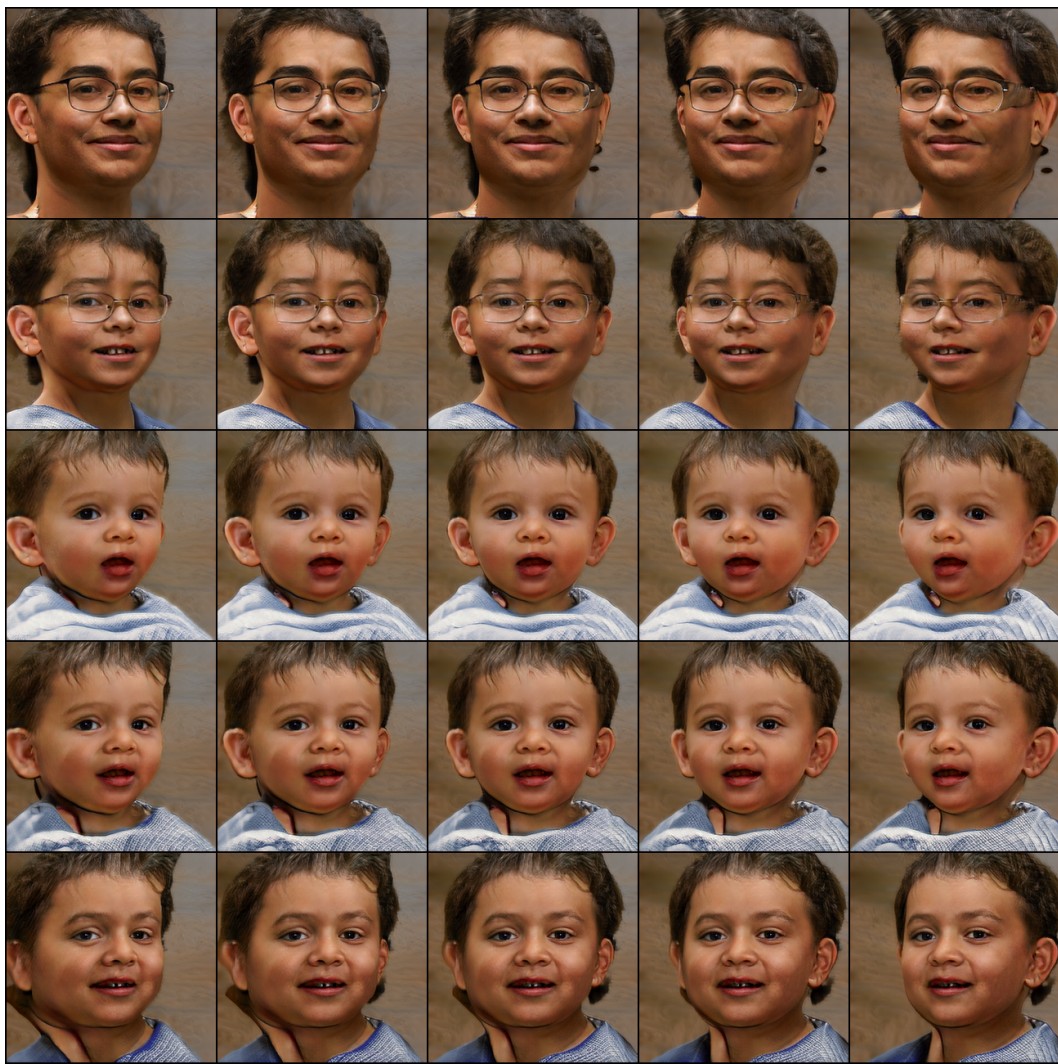

Figure 4: **Effect of Pose Conditioning**. From left to right we vary the pose used for rendering $\xi'$ and from top to bottom we vary the pose used for conditioning $\xi$. All samples are generated using the same latent $\mathbf{z}$.

turn, the background models the body of the person which should be part of the foreground. Learning to disentangle background and foreground without direct supervision, e.g., instance masks, is often ambiguous which makes it a challenging task. The middle column in Fig. 2 displays a common failure case of our model on AFHQ: The whiskers of the cat are connected to its body. This is likely a consequence from the depth variance and total variation regularization we apply to obtain a single sharp surface and compact geometry. The right column in Fig. 2 shows an occasional failure on FFHQ. For some samples, we observe that the hair is directed inward to the head instead of outward. In the rendered image this effect is not visible which suggests that it likely results from ambiguity in the training data.

**Uncurated Samples.** We provide additional samples of our method for FFHQ [14] in Fig. 5, AFHQ [5] in Fig. 6, and Carla [19] in Fig. 7.

**Quantitative Comparison.** As FID is biased towards the number of images [2], we explicitly annotate the number of generated images for the results reported in Table 2 of the main paper in Table 1. Most works evaluate FID either using 20k or 50k generated images. We evaluate our method

for both cases. For our method and the considered datasets, the difference between using 20k or 50k generated images is reasonably small.

# 4   Societal Impact

This work considers the task of generating photorealistic renderings of scenes with data-driven approaches which has potential downstream applications in virtual reality, augmented reality, gaming and simulation. While many use-cases are possible, we believe that in the long run this line of research could support designers in creating renderings of 3D models more efficiently. However, generating photorealistic 3D-scenarios also bears the risk of manipulation, e.g., by creating edited imagery of real people. Further, like all data-driven approaches, our method is susceptible to biases in the training data. Such biases can, e.g., result in a lack of diversity for the generated faces and have to be addressed before using this work for any downstream applications.

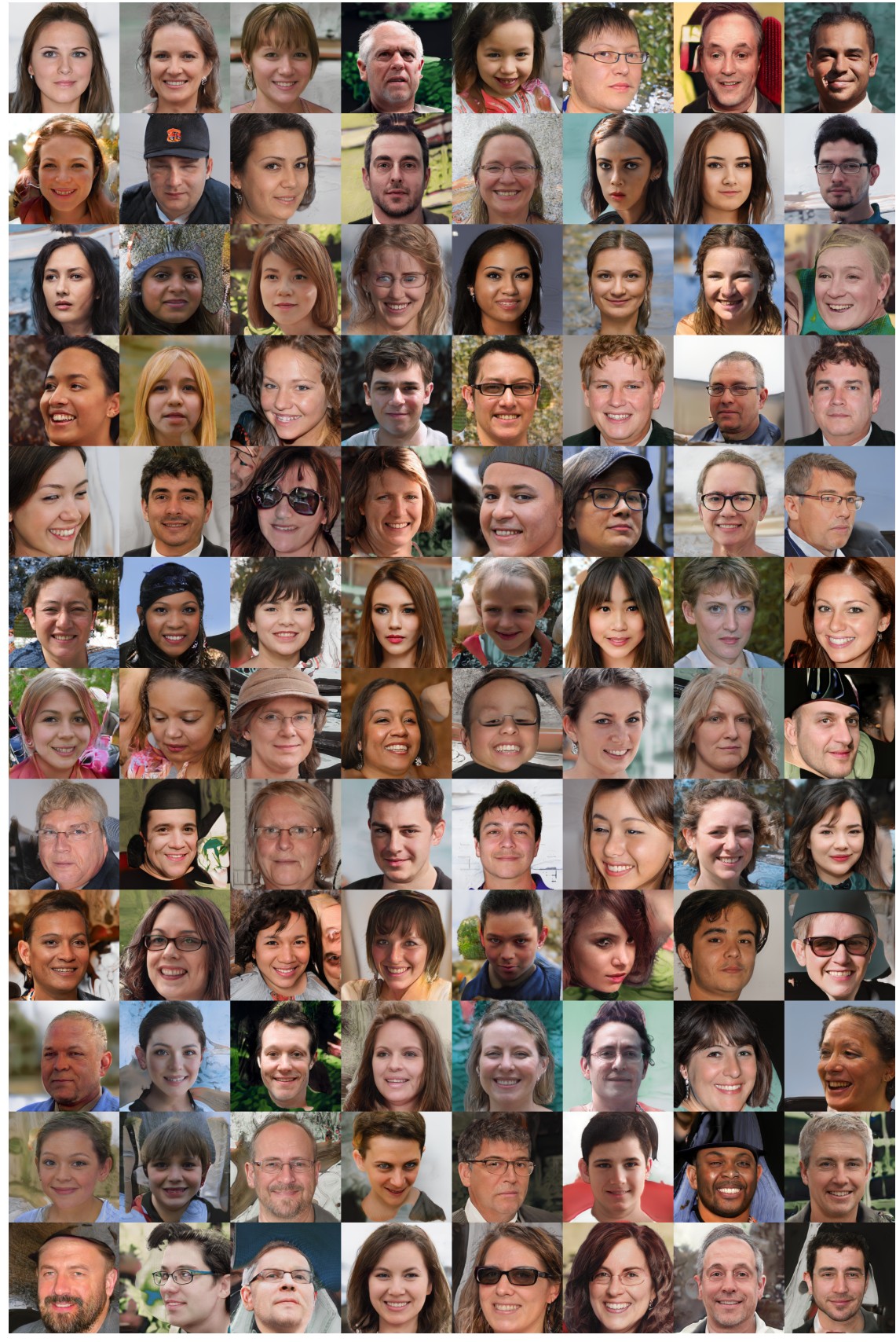

Figure 5: **Uncurated Samples for FFHQ [14].** We use truncation with $\psi = 0.7$.

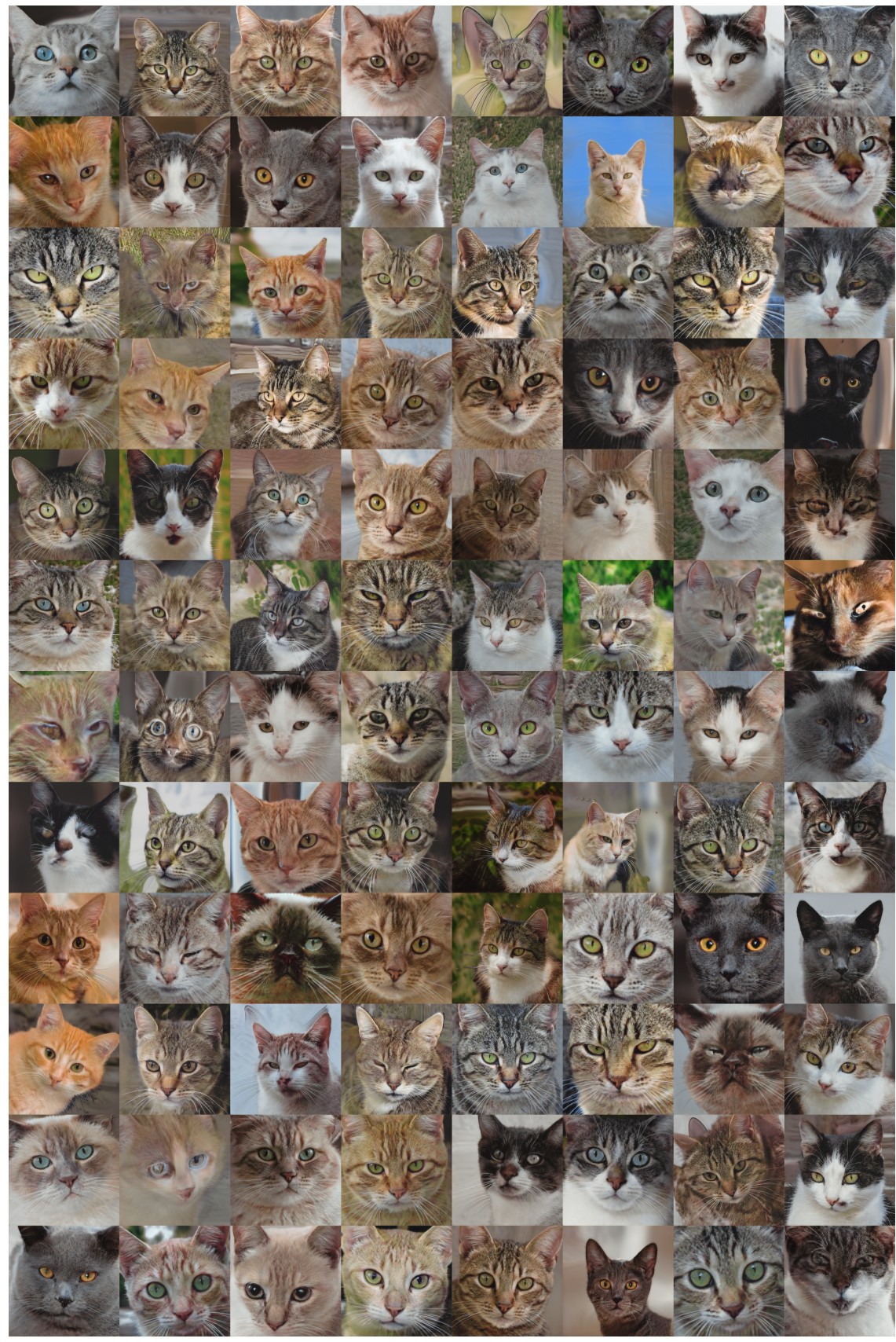

Figure 6: **Uncurated Samples for AFHQ [5].** We use truncation with $\psi = 0.7$.

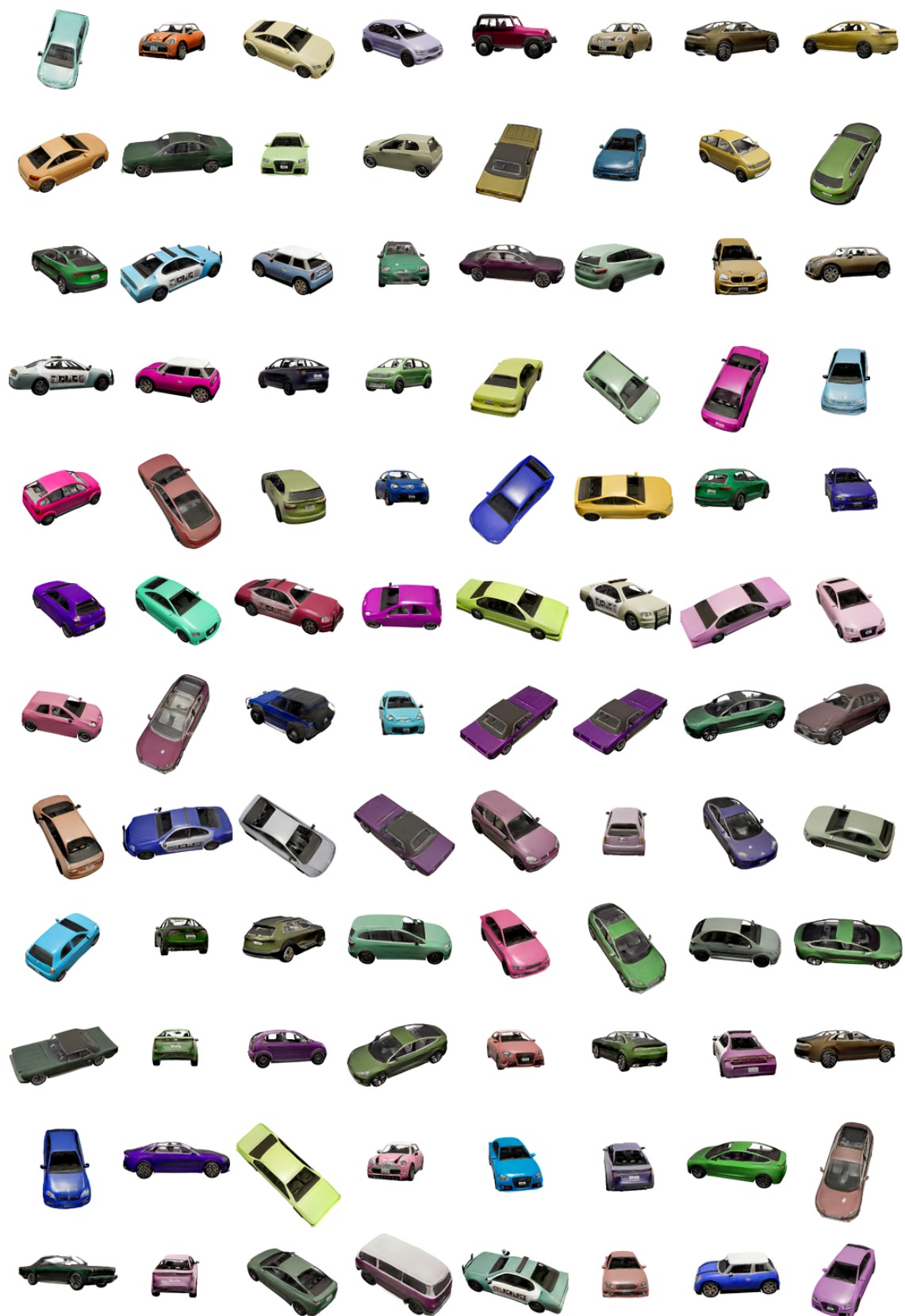

Figure 7: **Uncurated Samples for Carla [19].** We use truncation with $\psi = 0.7$.

## Acknowledgments and Disclosure of Funding

We acknowledge the financial support by the BMWi in the project KI Delta Learning (project number 19A19013O), the support from the BMBF through the Tuebingen AI Center (FKZ:01IS18039A), and the support of the DFG under Germany's Excellence Strategy (EXC number 2064/1 - Project number 390727645). Andreas Geiger and Michael Niemeyer were supported by the ERC Starting Grant LEGO-3D (850533). We thank the International Max Planck Research School for Intelligent Systems (IMPRS-IS) for supporting Katja Schwarz and Michael Niemeyer. This work was supported by an NVIDIA research gift. We thank Christian Reiser for the helpful discussions and suggestions. Lastly, we would like to thank Nicolas Guenther for his general support.