# OpenReview forum: "VoxGRAF: Fast 3D-Aware Image Synthesis with Sparse Voxel Grids"
_NeurIPS.cc/2022/Conference — NeurIPS 2022 Accept_

### Official Review · Reviewer_2xja · 2022-06-14

**Rating:** 6
**Confidence:** 4
**Soundness:** 3 good
**Presentation:** 3 good
**Contribution:** 3 good

**Summary:**

The paper proposes a novel 3D GAN architecture for rendering of 3D consistent objects supervised by collection of single-view images. The key innovation is the introduction of sparse voxel grid as a more computationally efficient representation when compared to the most commonly used MLPs. The authors describe how sparsity is enforced and used to reduce memory constraints of their approach. As a result, they can render medium-sized images from their voxel grid directly without need to explicit upsampling. The validation shows the method produces better images at lower computation cost than previous work and it is in some respects competitive with more recent methods published in parallel.

**Questions:**

11) If voxels are pruned based on visibility in "a" rendered view (L167), cannot they be later missing after occlusion changes due to a new viewpoint at test time?


**Limitations:**

12) The authors have mentioned the performance may drop is scene is more complex and pruning is not very effective. Other limitations I suspect is the lack of geometric smoothness in extracted shapes and the high training cost.


**Strengths And Weaknesses:**


[Originality]

1) The method combines an established concept of NeRF-based 3D-GAN with previously described sparse voxel convolutions. The authors further describe several key tricks that are required for successful and efficient training. Many of them are inspired by similar work, but at least the loss L_D seems unique to this publication. Overall, the technical contribution is incremental but, nevertheless, it leads to a significant improvement upon SOTA in the target application (here I ignore the parallel work).


[Clarity]

2) The exposition is very clear, easy to read and I have not discovered many typos or grammar issues.

3) I find problematic the way how parallel work is discussed. GRAM, StyleNERF, and EG3D are all parallel work so they do not count against contributions of this paper. I appreciate that the authors tacked the task of discussing their properties and that they included them in their result tables. However, I am not entirely sure all conclusions that they have made are fully substantiated. For instance, L264 seem to suggest that EG3D is less view consistent than the proposed method but no such effect is shown. Both methods use internal 128px representation for the radiance field and therefore, any further processing increasing the output image resolution is in principle similar for both. The authors also argue, that the advantage of their method is the possibility of faster rendering after the first frame (Table 3). However, the same could be said about EG3D as it also could precompute the StyleGAN representation and then just sample the features the same way the new paper does. In general, I find these over-extended assumptions about parallel work both not properly validated and also unnecessary. As these work are not counted as a previous work, there is no need to prove that the new method is in all or any respects better. The authors should rather focus on properties of their own work and show why it is excellent. Since the general architecture and performance appears quite similar, focusing on the possibilities of the explicit voxel representation seem like a clear goal.

[Validation]

4) The method is evaluated on common datasets and the results are clearly better than previous work and comparable to parallel work. The results are limited to 256px resolution with is also comparable to recent methods.

5) There are no quantitative experiments done to validate the claim of better geometry and 3D consistency (L74). The authors could also attempt to measure depth and pose accuracy as done in previous work [4]. Since there is still a small CNN used for the image refinement it does not seem obvious that the method is perfectly view consistent.

6) The extracted shapes appear quite noisy (Fig. 5).

7) The authors only ablate the loss L_DV and only in terms of sparsity. Neither of the losses that are used for regularization (there are 4 in total) is evaluated in terms of end-to-end performance impact.

8) The training is extremely costly even compared to MLP-based NeRF GANs (L224).


[Significance]

9) The paper tackles a relevant problem of 3D representation generation and it does take a path distinct from other work. While it is not clear whether the method actually is clearly better than some of the parallel work, it does clearly beat the previous baselines. The voxel representation may have practical benefits that have not been mentioned in the paper. E.g. I can imagine that they would be easier to edit in a 3D modeler. However, no such application has been shown.


[Conclusion]

10) While I see some issues with the exposition, the paper is of a high quality overall. The discussion of the performance and relation to other work can be addressed. The bigger concern is the small scope of the validation. While the most critical image synthesis quality is directly evaluated on common datasets, this leads to a total of three numbers reported in Table 2. The 3D consistency, design ablation (with exception of Table 1) and the shape quality are not directly validated in a quantifiable way. That being said, I do not suspect such experiments to significantly change the narrative of the paper so I am still inclined towards acceptance, especially if the authors can improve upon these aspects in the final version.

---

> ### Author Response · Authors · 2022-08-02
> **Response to Reviewer 2xja**
>
> Thank you for your constructive comments and valuable concerns.
>
> > For instance, L264 seem to suggest that EG3D is less view consistent than the proposed method but no such effect is shown.
>
> We agree and will replace the statement with:
> “This is expected as the neural renderer adds flexibility while sacrificing 3D-consistency, see Fig. 4.” -> “This is expected as a neural renderer can add flexibility. But, as shown in Fig. 4, for StyleNeRF it reduces 3D-consistency.”
>
> > The authors also argue, that the advantage of their method is the possibility of faster rendering after the first frame (Table 3). However, the same could be said about EG3D as it also could precompute the StyleGAN representation and then just sample the features the same way the new paper does.
>
> We agree that this is possible but remark that EG3D still needs to densely query the 3D representation. For caching the tri-plane features, EG3D reports a speed up from 27FPS to 36FPS at resolution 256^2 (Table 3 in their paper). Instead, our approach only queries 3D space sparsely which allows rendering at 167FPS.
>
> > There are no quantitative experiments done to validate the claim of better geometry and 3D consistency (L74). The authors could also attempt to measure depth and pose accuracy as done in previous work [4]. Since there is still a small CNN used for the image refinement it does not seem obvious that the method is perfectly view consistent.
>
> The code for computing the metrics in [4] is not publicly available. For this rebuttal, we implemented our own version of the depth and pose metric following the description in [4]. Below, we report the results for our approach, GRAM, StyleNeRF and EG3d for reference. We also report the standard deviation across the 1024 samples used for evaluation. While the results agree with our qualitative analysis of view consistency in Figure 4 and the supplementary video, we find that both metrics are very sensitive to the latent code and the sampled poses, as indicated by the large standard deviations. As no established evaluation pipeline exists, our results should not be directly compared to the numbers in [4] as the implementation and pose sampling might differ. We will add this experiment to the supplementary.
> |                | Depth ↓       | Pose ↓                    |
> |-----------|-----------------|---------------------------|
> | StyleNeRF | --                 | 0.051 ± 0.047         |
> | GRAM    | 0.48 ± 0.24 | 0.013 ± 0.013         |
> | EG3D     | 0.29 ± 0.30 | 0.0018 ± 0.0031     |
> | VoxGRAF   | 0.33 ± 0.23 | 0.00045 ± 0.00079 |
> Note that for StyleNeRF depth can only be rendered at resolution 32^2 and we thus omit evaluating the Depth metric for it.
>
> > The authors only ablate the loss L_DV and only in terms of sparsity. Neither of the losses that are used for regularization (there are 4 in total) is evaluated in terms of end-to-end performance impact.
>
> We ablate the performance of all four regularizers below on models trained on FFHQ with R_I=128 and R_G=64. The regularizers do not significantly impact end-to-end performance measured in FID but help to stabilize training (L.208). We will add this ablation to the supplementary.
> |                                       | FID  |
> |---------------------------------------|------|
> | full                                  | 14.2 |
> | w/o L_DV                    | 14.8 |
> | w/o L _TV                  | 15.1 |
> | w/o L_cvg^fg    | 14.6 |
> | w/o L_cvg^bg     | 14.8 |
>
> > The training is extremely costly even compared to MLP-based NeRF GANs (L224).
>
> Our method’s training time (3-8d) is comparable to EG3D (8.5d), StyleNeRF(3d) and GRAM (3-7d), all reported on 8 Tesla V100 GPUs.
> > If voxels are pruned based on visibility in "a" rendered view (L167), cannot they be later missing after occlusion changes due to a new viewpoint at test time?
>
> We only prune voxels based on the rendered view for training. At inference, we prune voxels solely based on their density to amortize the rendering costs per scene. We find that this does not visibly affect the rendered images. We will clarify this in the paper.

---

> > ### Comment · Reviewer_2xja · 2022-08-03
> > **Response to rebuttal**
> >
> > I thank the authors for the rebuttal. It has largely answered all my concerns. I just want to ask some clarifying questions and also get some more insight into the issues raised by the other reviewers.
> >
> > 1) The newly provided depth/pose numbers for the other methods were obtained by extracting their results and applying your own implementation of the metrics to all of them? I am just checking that they are mutually comparable.
> >
> > 2) In the paper you strongly emphasize that EG3D and GRAM are concurrent works which I took for granted but after seeing the other reviews, I realized that CVPR papers were already accepted long before the NeurIPS submission. Perhaps it would be fitting to tone this down a bit. Also please update the references to include the venue instead of arxiv (that applies more broadly).
> >
> > 3) Regarding the Minkowski convolution (Reviewer eE7o) - Do I understand it correctly that in the revised version you will only use dense convolutions for the 256^3 grid size? Does that mean that you tested Minkowski at R_G=256 and it does provide a performance benefit over dense convolution there? However, it provides no benefit at R_G=128 resolution? Can you please point me to the experiment with R_G > 128?
> >
> > 4) Regarding the pose conditioning of the generator (Reviewer eE7o): I still do not see why is the generator conditioned by the pose. Why should the pose change the density distribution? I see a similar conditioning of the triplane in EG3D. Does this follow the same reasoning? Thank you for the test-time example figure. Is it possible to also provide a comparison of *training* with and without this feature in the final version (can be in the supplement)?

---

> > > ### Author Response · Authors · 2022-08-03
> > > **Response**
> > >
> > > Thank you for the quick reply to our rebuttal.
> > >
> > > 1. Yes, exactly. We ran all methods using our implementation of the depth and pose metric, so results are mutually comparable. The evaluation of methods using our own implementation is also why our reported numbers for EG3D vary slightly from the numbers reported in their paper (Depth: 0.29(ours) vs 0.31 and Pose: 0.0018(ours) vs 0.005).
> > >
> > > 2. Thanks for the comment, we will adjust the tone accordingly. Please note, however, that at date of submission, only arxiv preprints of the respective projects were available and no code was released, hence a direct comparison was not possible.
> > >
> > > 3. Yes, after the submission deadline we found that at R_G=128^3 Minkowski’s sparse convolution provides no speed up over PyTorch’s dense convolution for our generator architecture. We did not train a model with R_G=256^3, though. Whether the sparse or dense implementation is faster does not only depend on the sparsity but also on the number of channels and on the resolution.
> > > We hence believe that both, a variant using Minkowski’s sparse implementation and a variant with PyTorch’s dense implementation can be valuable to the community. We will discuss both implementations in the camera ready version and include both implementations in our code release.
> > > Note that this is only a change in implementation and using the dense implementation will not change that the generated grids are sparse, as we still have to prune them for efficient volume rendering.
> > >
> > > 4. The purpose of the pose conditioning is to model biases in the training data. For example, in FFHQ people tend to smile with an open mouth more often when seen from the front than from the side. Hence, whether a person smiles with an open mouth is correlated to the camera pose in the training data. As this will also change the geometry, the density prediction is also conditioned on the pose. Another example is that children are photographed mostly from the front. Hence, we observe that conditioning our trained generator on a centered pose will often generate a child while conditioning on a camera pose on the side does not. This can also be seen in the pose conditioning ablation we added in the response to Reviewer eE7o (https://imgur.com/a/y7d4njy), where the middle row corresponds to the centered camera pose. For EG3D the reasoning for conditioning the generator is the same.
> > > We will follow your suggestion and include a comparison for training without pose conditioning to the final version.

---

> > > > ### Comment · Reviewer_2xja · 2022-08-04
> > > > **Response**
> > > >
> > > > Thank you, that answers my questions.

---

### Official Review · Reviewer_xKgP · 2022-06-21

**Rating:** 5
**Confidence:** 3
**Soundness:** 3 good
**Presentation:** 3 good
**Contribution:** 1 poor

**Summary:**

Authors propose an approach to generate high quality images from a latent code that are multi-view consistent. The intuition is to replace the MLP that is used to parameterized radiance fields, with a hybrid approach that relies on sparse 3D convolutions to speed up the rendering time. Additionally, they propose a way to disentangle the foreground and background.

**Questions:**

The main question is around novelty and experiments/comparisons. I'd like authors to convince me that this paper introduces some different capability compared to [4] or that they solve the problem better. In the current form, I do not think the paper is doing a good job at convincing the reader that this method is pushing the state of art.

---
Post rebuttal: Authors made the fair point that [4] was concurrent work and there was no code release at the time of the submission. Hence I am leaning towards acceptance.

**Limitations:**

Yes, discussed.

**Strengths And Weaknesses:**

++ Relevant problem for the community. Generating high quality renderings with an adversarial approach is an unsolved problem despite the recent progress

++ Clarity: the paper is overall clear, authors put lots of effort to describe every details of the approach. They sufficiently discuss limitations and ethical concerns.


-- Novelty, Contributions and Experiments. Whereas the general idea is interesting, the first thing that comes to mind once reading this paper is: why is this approach better than EG3D [4]? Indeed, they tackle the same problem and I would argue that the results of EG3D are actually better. Quantitative comparisons that author report seem to be aligned with this, and qualitative experiments are not showed (I would encourage authors to report those). According to authors, the main difference between their approach is explained at l142-l143, which is "In contrast to recent works, we do not use a coordinate-based MLP..". While this is true, it is not demonstrated that the proposed method is actually better.

---

> ### Author Response · Authors · 2022-08-02
> **Response to Reviewer xKgP**
>
> Thank you for reading and reviewing our work. We appreciate your feedback and address your concerns below.
>
> > Why is this approach better than EG3D [4]? (...) Quantitative comparisons that author report seem to be aligned with this, and qualitative experiments are not showed (I would encourage authors to report those).
>
> We remark that [4] was developed concurrently to our work and at the time of submission no code of [4] was publicly available. We contacted the authors beforehand, but unfortunately we could not get access to preliminary code. Hence, it was not possible to add [4] to the qualitative comparisons in the paper but we added quantitative results from [4] to compare with them to the best of our abilities.
>
> > I'd like authors to convince me that this paper introduces some different capability compared to [4] or that they solve the problem better. In the current form, I do not think the paper is doing a good job at convincing the reader that this method is pushing the state of art.
>
> - In contrast to [4] our model learns to disentangle background and foreground. This can be useful for many downstream applications, e.g. to integrate generated assets into new environments.
> - One major limitation of [4] is the slow volume rendering because querying 3D space densely is prohibitively costly (L.48). E.g. in [4] it is stated that increasing the rendering resolution from 64^2 to 128^2 (while keeping the remaining architecture fixed) almost doubles the time to train on 1000 images from 24s to 46s. In contrast, our approach queries 3D space sparsely which makes volume rendering more efficient.
> - Due to EG3D’s slow volume rendering, [4] relies on a neural render (2D CNN) to synthesize images which may entangle viewpoint and 3D content (L.35) and requires additional regularization to reduce this effect. Instead, our approach generates a sparse representation that can be rendered efficiently without a neural renderer.
> We argue that generating 3D content at high resolution directly is desirable for many downstream applications, e.g. for integrating assets into physics engines (L.53). Our approach improves image fidelity over state-of-the-art methods without neural rendering and reduces the gap to models that build heavily on neural rendering (Table2, L.285).
> - Sparse voxel grids further allow to amortize the rendering costs per scene, s.t. our approach can render novel views of a generated instance at 167FPS at image resolution 256^2. We remark that [4] can cache the tri-plane features, i.e. the first part of the forward pass. As shown in [4], this increases rendering speed from 27FPS to 36FPS while being run on faster hardware than our model.
>
> We would like to reiterate that at date of submission no code of [4] was publicly available and hence an in-depth comparison was not possible. We compared to the best of our abilities by including quantitative results from their paper in our work. We hope that our arguments are able to resolve your concerns.

---

> > ### Comment · Reviewer_xKgP · 2022-08-08
> > **Thanks**
> >
> > I appreciate the authors' rebuttal. Since EG3D was indeed a concurrent work, I think it is fair to not consider it in this review. Hence I am leaning towards acceptance.

---

### Official Review · Reviewer_eE7o · 2022-07-12

**Rating:** 7
**Confidence:** 3
**Soundness:** 4 excellent
**Presentation:** 4 excellent
**Contribution:** 4 excellent

**Summary:**

The paper introduces a 3D-aware image generation network that usess a Plenoxels-style voxel grid [1] as its scene representation. The complete network architecture consists of a sparse 3D CNN generator with novel upsampling layers that ensure sparsity of the upsampled densities. The full resolution voxel grid has size $128^3$ and stores RGB+density values for the occupied voxels.

In a parallel branch, there is a 2D GAN (following StyleGAN v2) that generates the scene background.

The sparse voxel grid is rendered at the required output resolutoin using conventional volume rendering techniques. The foreground is post-processed by a shallow (two hidden layers) refinement CNN, whose output is then composited on the the background image.

At training time, the resulting image is passed to a discriminator based on StyleGAN2. The authors describe several regularizing losses, including a novel sparsity term on the voxel densities.

The authors report both quantitative and qualitative comparisons to state-of-the-art 3d-aware GANs and demonstrate results that demonstrate a high degree of geometric consistency across viewpoints.

[1] Yu et al. Plenoxels: Radiance Fields without Neural Networks. 2022

**Questions:**

 - See previous section for questions about voxel & generator view dependence.
 - What is the speedup gained from using sparse convolutions at grid sizes > $32^3$?

**Limitations:**

The authors are clear about the limitations of their work. Societal impacts are adequately discussed.

**Strengths And Weaknesses:**

This is a strong paper with an original technical contribution in the sparse CNN generator and the use of a sparse RGB-density voxel grid as the representation for a 3D-aware GAN. The voxel visualizations in the paper, as well as the animated sequences in the supplemental video, demonstrate the geometric plausibility of the generated representations.

The paper is very well written and the discussion of related works is thorough. The comparisons to related work are quite comprehensive, and a key ablation study is included.

A particularly exciting aspect of the method is the very fast rendering times (after a slower initial generation of the voxel grid), which could allow this technique to run at interactive framerates on low-powered platforms.


--

Even though the results exhibit high geometric consistency, the visual quality of the generated still frames does not quite match that of some other methods. This limitation is acknowledged in the paper.

My main critique is around the description of viewpoint dependence. Plenoxels stores spherical harmonic coefficients in the voxels, while the proposed method seems to store RGB values without directional dependence (Eq. 3.). This is a quite significant departure from other NeRF-inspired representations, and should be made more explicit in the discussion of the method (Section 3.1).

It is somewhat surprising that the foreground generator is conditioned on the camera pose $\xi$, since the volumetric grid can be rendered from arbitrary viewpoints $\xi'$, even those that it was not conditioned on. Could the authors clarify for which results in the paper $\xi = \xi'$ ? Would it be a viable ablation to compare two animated sequences, one with $\xi = \xi'$ at each frame, and one where the generator was invoked only once?

---

> ### Author Response · Authors · 2022-08-02
> **Response to Reviewer eE7o**
>
> Thank you for the valuable comments and feedback.
> > My main critique is around the description of viewpoint dependence. Plenoxels stores spherical harmonic coefficients in the voxels, while the proposed method seems to store RGB values without directional dependence (Eq. 3.). This is a quite significant departure from other NeRF-inspired representations, and should be made more explicit in the discussion of the method (Section 3.1).
>
> Our 3D generator is conditioned on the pose and thereby contains directional dependence. We will clarify this in L.156 by adding:
> “Note that in contrast to most existing coordinate-based 3D-aware GANs, see Eq. 1, we do not condition the 3D generator on the view direction (per-ray). Instead, we follow [4] and condition it on the pose ξ (per-image) to model directional dependencies.“
>
> > It is somewhat surprising that the foreground generator is conditioned on the camera pose ξ since the volumetric grid can be rendered from arbitrary viewpoints ξ′ ,even those that it was not conditioned on. Could the authors clarify for which results in the paper ξ=ξ′ ? Would it be a viable ablation to compare two animated sequences, one with ξ=ξ′ at each frame, and one where the generator was invoked only once?
>
> We sample the camera pose for conditioning ξ randomly for all results in the paper, i.e. ξ is in general different from ξ′ for all shown results. The suggested ablation is indeed interesting. The results provided via the link below illustrate that the pose conditioning is not only used to model slight changes, e.g. of the eyes or the smile, but can alter the general appearance of the generated instance. However, by fixing the pose conditioning during inference, view-consistent images can be generated. We will add this ablation to the supplementary.
>
> https://imgur.com/a/y7d4njy
>
> Figure 1: Effect of pose conditioning. From left to right we vary the pose used for rendering ξ′ and from top to bottom we vary the pose used for conditioning ξ. All samples are generated using the same latent z.
>
> > What is the speedup gained from using sparse convolutions at grid sizes > 32^3?
>
> After the submission deadline we realized that for our generator with voxel grids up to resolution 128^3, the sparse convolution implementation of the Minkowski library does not provide significant gains over PyTorch’s dense convolution (memory is slightly reduced but runtime increases). We will therefore add another variant of our generator to the camera ready version that replaces Minkowki’s sparse convolution with PyTorch’s dense convolution. Note that changing the implementation of the convolution operation will not change the sparsity of the generated voxel grids. A sparse representation remains key for fast volume rendering and can still only be obtained by pruning the grid based on the output of the previous resolution. Hence, the pruning operator ρ and the progressive growing of the model remain unchanged. We will add this ablation to Table 1 as changing the convolution implementation will mainly reduce the runtime for the generator forward pass t_G^fg+bg while it might slightly increase memory. We will further include a comparison on FID but expect that the results will be similar for both models as the computation itself does not change.
> Lastly, we remark that in initial experiments we indeed found Minkowski’s sparse convolution advantageous for runtime and memory. While changes to the generator architecture (e.g. lower channel dimension at higher resolution) neutralized these benefits, future work might still profit from sparse convolutions when going to larger grid resolutions and larger scenes. Hence, we consider both variants interesting to the reader and we will release our source code including both variants.

---

### Meta-Review · Area_Chair_4CzE · 2022-08-27

**Recommendation:** Accept
**Confidence:** Certain

**Metareview:**

It is valuable now to introduce this technical idea, even if the results do not quite match existing methods.  Future work building on this idea may well do so, and it would impede the progress of the subfield to demand both the new idea and SOTA results.

The rebuttal takes on extra work building on the reviewers' suggestions, which gives confidence that the final paper will be of high quality.

At the same time, the primary criterion for oral presentation is importance to the community as a whole, so the relatively narrow scope (3D computer vision) would really require more world-leading results, and possibly demonstrations of a wider set of applications in order to alert practitioners in adjacent subfields.



**Award:**

No

---

### Decision · Program_Chairs · 2022-09-14

Accept